# Experts on Demand: Dynamic Routing for Personalized Diffusion Models

## Abstract

Diffusion models have excelled in the realm of image generation, owing to their expansive parameter space. However, most users only exploit a fraction of the available capabilities for specialized image categories synthesis. These specific requirements for individual users are often persistently fixed over the long term, for example, a pet store pursues images of cats and dogs, which poses an efficiency challenge due to the computational complexity involved. In this paper, we introduce Mixture of Expert Diffusion Models (MoEDM), a personalized and efficient strategy for large-scale diffusion models specific to certain applications. By employing dynamic routing, MoEDM selectively activates only indispensable neurons, thereby optimizing runtime performance for specialized tasks while minimizing computational costs. Our MoEDM doubles the sampling speed without compromising efficacy across various applications. Moreover, MoEDM's modular design allows straightforward incorporation of state-of-the-art optimization methods such as DPM-Solver and Latent Diffusion. Empirical assessments, validated by FID score, KID score and human evaluation, confirm the advantages of MoEDM in terms of both efficiency and robustness.

## 1 Introduction

In the realm of machine learning, the allure of massive, versatile models often eclipses the practical considerations of resource constraints and application specificity. Leveraging state-of-the-art diffusion models like SDXL-1.0 (Podell et al., 2023) with their staggering 3.5 billion parameters to fulfill nearly any image generation requirements may seem like the ideal strategy. However, this approach often proves to be a computational quagmire. The issue is exacerbated in real-world deployments, as these behemoth models struggle with efficient sampling, thereby magnifying the need for resource-optimized solutions tailored to specific application contexts. Despite the advent of various optimization techniques, including fast diffusion samplers like DPM-Solver (Lu et al., 2022) and lightweight diffusion architectures like Latent Diffusion (Rombach et al., 2022), there still exists considerable room for compressing model parameters. This compression potential stems from a primary challenge that remains unaddressed: the customization of models to meet user-specific requirements.

Do users truly need such a comprehensive but huge model? As described in Figure 1, in specialized use-cases like rendering images of cats and dogs for a pet store all along, deploying a general-purpose diffusion model is not just inefficient but severely wasteful. This excess in model complexity does more than consume valuable computational resources; it also undermines the efficiency of sampling procedures. Therefore, there is a compelling case for crafting streamlined, purpose-built models that maximize computational efficiency without sacrificing utility. We believe that small-scale customized models, instead of large-scale generalized models, are undoubtedly the future path to popularizing artificial intelligence, given the unattainability of large-scale computational resources and the instability of cloud servers. This underpins our development of compact and computationally efficient diffusion models.

Navigating the path to this optimal blend of efficiency and functionality is complex, fraught with hurdles including the inherent time-dependent complexities associated with diffusion models. While some existing efficiency-centric solutions, such as parameter pruning (Liu et al., 2018), offer static but partial relief, these

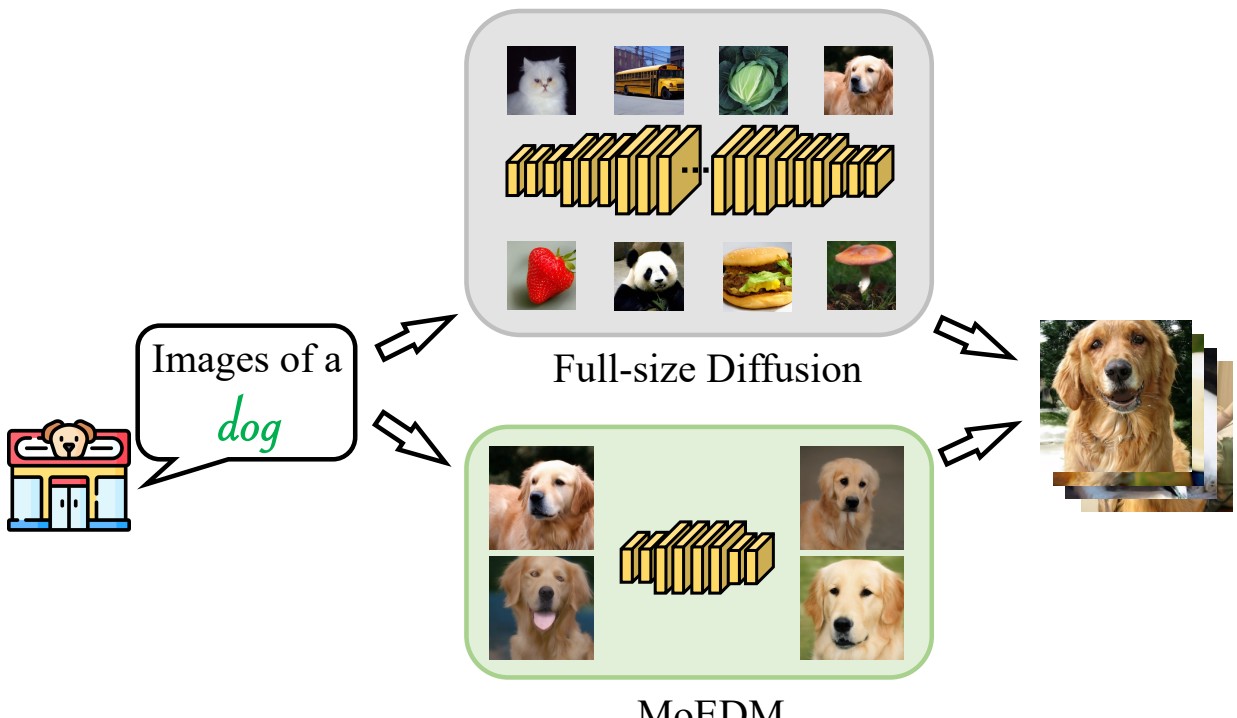

Figure 1: Teaser of MoEDM, and its comparison with traditional full-size diffusion models. The substantial number of parameters designed to accommodate various functions is inefficient for users who maintain a long-term, specific demand, such as a pet store's requirement for images of animals.

are generally tailored for feed-forward architectures and often fall short in preserving the performance attributes of diffusion models. Furthermore, Parameter-Efficient Fine-Tuning (PEFT) offers swift personalized fine-tuning (Zaken et al., 2021; Hu et al., 2021), however it remains unable to shed the excessive computational load in the sampling process stemming from a multitude of parameters. Meanwhile, DreamBooth (Ruiz et al., 2022), serving as a classical personalized diffusion model, offers a swift option for personalized customization of diffusion models. However, it still grapples with a crucial issue—during the sampling stage, the generation of an image still necessitates the collective involvement of all parameters in the entire model, making it a time-consuming process.

In this study, we present Mixture of Expert Diffusion Models (MoEDM), a resource-efficient methodology for users' personalized needs with minimal computational cost. Leveraging the concept of dynamic routing, as established in prior research (Han et al., 2021), MoEDM enhances the efficacy of task-centric diffusion models by judiciously activating pertinent neurons. Initially, our approach involves the identification and removal of non-essential parameters, thereby streamlining the model for a designated task. The remaining neurons are then expanded into super-layers through a *training-free* gated strategy. Finally, the model will be fine-tuned through task-specific image datasets. During the sampling stage, only a singular pathway is engaged, substantially lowering computational demands. Through this approach, MoEDM offers a nuanced yet efficient solution for developing task-specific diffusion models that maintain robust performance with significantly higher computational efficiency in users' specific tasks, at the same time incurring acceptable low training costs.

MoEDM yields considerable advantages, notably a 100% enhencement in sampling speed, owing to a significant reduction of active parameters. This enhancement is accomplished without sacrificing model efficacy, corroborated by empirical evaluations on ImageNet (Deng et al., 2009), FFHQ (Karras et al., 2017) and text-to-image synthesis. Specifically, we employ MoEDM in a range of applications—from subset creation in ImageNet and domain adaptation to FFHQ, to text-to-image synthesis—consistently realizing gains in sam-

pling efficiency without any trade-off in quality. By focusing on the dynamic architectures, we can minimize computational overhead without sacrificing either versatility or reliability, which is crucial for engineering lean yet robust, application-specific diffusion models.

The main contributions of our study are summarized as follows:

- We present MoEDM, a personalized algorithm that simultaneously minimizes computational burden and expedites the sampling procedure in diffusion models. Crucially, MoEDM enhances the inference efficiency for designated tasks, maintaining intact the task-specific performance metrics.

- Through judiciously crafted ablation tests, we validate the robustness and adaptability of our methodology. These examinations further elucidate the intrinsic advantages conferred by our computational simplifications, especially in the realm of personalized applications.

- By drilling down into model parameters, MoEDM opens a new frontier for navigating the intricate landscape of diffusion model deployment. Its versatility is evidenced by seamless integration with existing, user-friendly models like DPM-Solver (Lu et al., 2022) and Latent Diffusion (Rombach et al., 2022), underscoring its exceptional scalability and cooperative efficacy.

## 2 Related Work

**Diffusion Models**  Diffusion models have gained significant traction as robust generative tools (Ho et al., 2020; Rombach et al., 2022; Dhariwal & Nichol, 2021; Nichol & Dhariwal, 2021; Kumari et al., 2022). Notwithstanding their prowess, the computational burden imposed by their extensive parameter sets renders them challenging to deploy in real-world applications. To mitigate these shortcomings, fast sampling methods like DDIM (Song et al., 2020) and DPM-Solver (Lu et al., 2022) aim to optimize the inference steps, thus accelerating the reverse denoising process. Latent Diffusion (Rombach et al., 2022) adopts the approach by relocating the diffusion process to the latent space, resulting in a more lightweight and efficient model.

These strategies are aimed at generalized diffusion models, detached from users' long-term specific requirements. Meanwhile, some personalized diffusion models (Ruiz et al., 2022; Zaken et al., 2021; Hu et al., 2021) do provide customization options for users, yet they still cannot escape the large-scale computational demand during the sampling phase. In this paper, we propose to squeeze diffusion models, addressing the core challenge of excessive parameterization. This parameter-level adaptation not only alleviates the computational burden but also resonates with specific user requirements that are often overlooked by existing solutions.

**Network Pruning**  In accordance with our research objectives, various techniques have been devised to optimize lightweight neural network models, specifically model pruning (Han et al., 2015; Molchanov et al., 2016; He et al., 2017; Liu et al., 2021). The essence of model pruning lies in identifying and eliminating parameters that have a minimal influence on model performance. For instance, value-based methods assess parameter significance through their numerical magnitudes (Han et al., 2015). In contrast, gradient-based methods evaluate parameter importance by examining associated gradient values (Liu et al., 2021). While these techniques demonstrate substantial efficacy in optimizing single-step feed-forward neural networks, they are less applicable to diffusion models. Diffusion models introduce unique computational intricacies (Li et al., 2023); they require multiple iterations of the same neural network model across sequential time steps, classifying them as recurrent or multi-step models. Consequently, there is still a dearth of straightforward and viable lightweight optimization techniques in the domain of diffusion models. Still focusing on generic diffusion models, the optimization strength of existing works (Kim et al., 2023; Yang et al., 2023) cannot achieve a qualitative leap.

**Dynamic Models**  Conventional model pruning techniques primarily focus on the irrevocable elimination of parameters, resulting in an unalterable decline in model performance. In contrast, dynamic models (Han et al., 2021; Wang et al., 2018; Lin et al., 2017; Liu & Deng, 2018) present a flexible architecture, adapting in real-time based on the input data. Pioneering works in this domain include Wide-DeepMoE (Wang et al.,

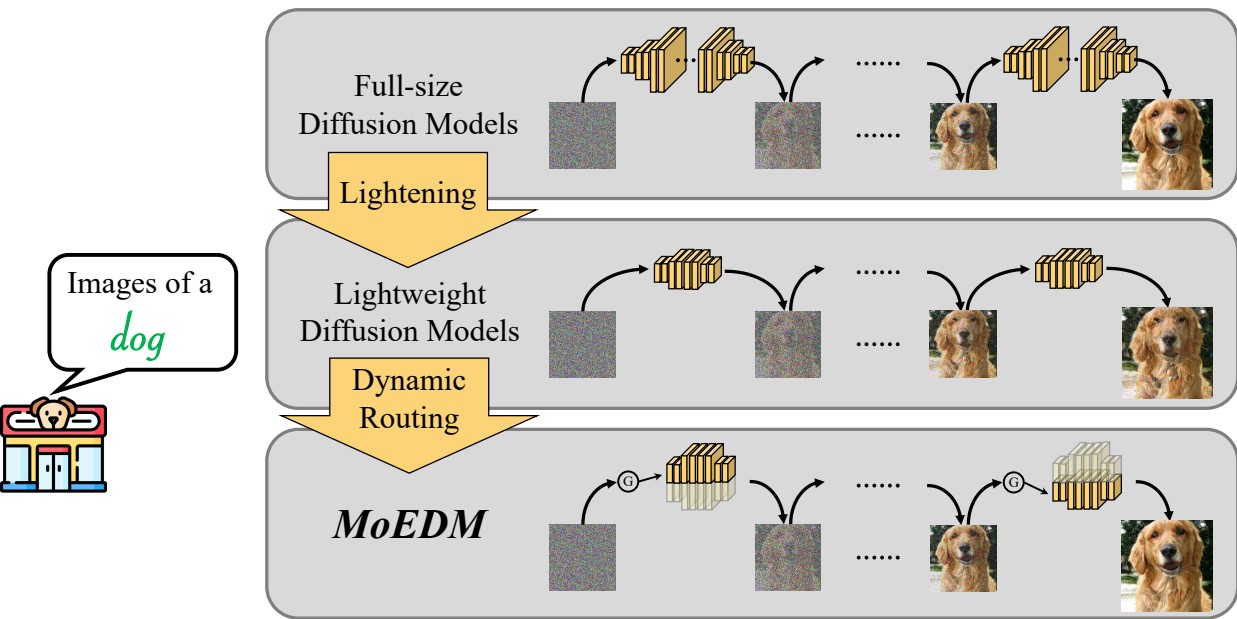

Figure 2: Overview of MoEDM, and its comparison with full-size models and lightweight models w/o dynamic. Neither full-size models nor naive-lightweight models can simultaneously achieve both high-quality image generation and faster sampling speed. With dynamic routing mechanism, MoEDM delivers a 2× sampling speed increase while maintaining the generation of high-quality images.

2020), which elevates performance metrics through both parameter expansion and customized parameter sets tailored for different input samples. This paradigm aligns with the principles of dynamic routing (Han et al., 2021; Cai et al., 2021; Li et al., 2020), thereby optimizing performance without inflating the parameter count for each computational pass. Diffusion models naturally fit into the framework of dynamic models; they execute diverse computations contingent on the temporal sequence of input samples. However, there remains a lacuna concerning the integration of the Mixture of Experts (MoE) approach within diffusion models. Existing attempts (Balaji et al., 2022; Podell et al., 2023) often deploy a rigid architecture, effectively rendering the entire model as a singular "expert". To address this limitation, our work draws inspiration from Wide-DeepMoE and augments the efficiency of lightweight diffusion models by morphing them into dynamic routing architectures at the layer level.

## 3 Method

Our approach, Mixture of Expert Diffusion Models (MoEDM), depicted in Figure 2, leverages tailored resource-efficient models to expedite sampling for each user's specific task. To sustain performance while reducing computational overhead, we introduce dynamic routing strategies into the model architecture. The remainder of this section elucidates our design philosophy and the underlying algorithm.

### 3.1 Parameter Scoring

To prune less essential parameters tailored to a specific task, we employ a scoring mechanism to evaluate the importance of parameters within an already well-trained model, focusing particularly on the convolutional layers, which constitute approximately 80% of the model's parameters. Notably, the layers at the UNet's extremities have fewer parameters compared to those in the middle (*e.g.*, in Guided Diffusion (Dhariwal & Nichol, 2021), 256 channels *vs* 1024 channels in a layer). We refer to the layers having the highest number of channels as "mid-layers" throughout this paper. Intuitively speaking, parameters at extremities interface directly with input and output images, suggesting their potential criticality. On the other hand, mid-layers, despite their expansive parameter space, are more susceptible to containing redundant elements. Several

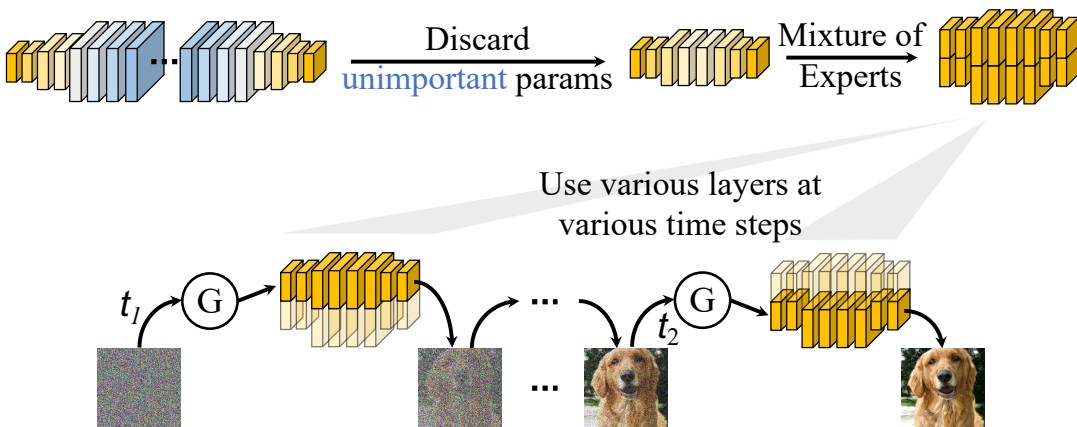

Figure 3: Method overview. MoEDM first discards mid-layers which is unimportant but contains the most channels. Next, we expand the remaining layers into "Mixture of Experts". During the sampling process, a training-free gated strategy directly activate an "expert" based on the input time step, $t$.

conventional techniques optimized for feed-forward models, like value-based (Han et al., 2015) and gradient-based (Liu et al., 2021) algorithms, yield sub-optimal results on diffusion models, and the distribution of pruned parameters diverged significantly from our initial hypothesis.

Consequently, we turn to a rudimentary yet effective scoring metric, $\mathcal{S}_c$, to better understand parameter significance within diffusion models. For each channel in a diffusion model, $c$, we remove it and observe the ensuing effects. We define $I_\theta(\epsilon, g)$ as the generation of images from Gaussian noise $\epsilon$ under a particular guidance $g$ using a pre-trained diffusion model, and $I_{\theta'_c}(\epsilon, g)$ as that using the same model with channel, $c$, removed. Here, $g$ can be various forms of guidance, such as category labels or text prompts. In this case, $\mathcal{S}_c$ is defined as follows:

$$\mathcal{S}_c := \|I_\theta(\epsilon, \ g) - I_{\theta'_c}(\epsilon, \ g)\|_1 \qquad (1)$$

A higher value of $\mathcal{S}_c$ signifies that omitting channel $c$ substantially influences the model's output, thereby serving as a reliable indicator for parameter significance. Empirical evidence further strengthens our claim: regardless of the discard ratio or the guidance applied, more than 90% of the parameters flagged for elimination belong to the middle layers of the network. This observation establishes a robust foundation and evidential support for the techniques proposed in our framework, hereby referred to as MoEDM. For detailed experimental results, please refer to Table 4 and Figure 6 in the Appendix Section A.

## 3.2 Mixture of Expert Diffusion Models

**Discard Layers** Lightweighting the models at the channel level offers precision but also introduces additional computational overhead due to the integration of group normalization layers and complex inter-layer connections. Consequently, we introduce a targeted strategy to discard superfluous parameters, namely discarding parameters at the layer level. In accordance with the conclusions drawn in Section 3.1, for those mid-layers holding the most parameters (e.g. 1024 channels each layer in Guided Diffusion), we directly discard them. Our results affirm that this layer-pruning approach leads to substantial gains in sampling speed without irreparable crash in model performance. Thus, the strategy effectively navigates the trade-off between efficiency and capability, establishing it as a feasible optimization technique for diffusion models.

**MoEDM** We propose the integration of "Mixture of Experts" (MOE) (Wang et al., 2020) into the existing diffusion models to maintain their performance. We leverage layer discarding and expand just the remaining layers, rather than the entire lightweight model like SDXL (Podell et al., 2023). This introduces greater

flexibility into the system, evident in our vector $\mathcal{G}$, which adjusts based on the time step $t$. Our formula for the output of layer $i$, $\mathcal{O}_i$, is therefore as follows:

$$\mathcal{O}_i = \mathcal{G}(t)_{i,\kappa} \cdot \mathcal{O}_{i,\kappa}(x, \ t, \ g)|_{\mathcal{G}(t)_{i,\kappa}=1, \mathcal{G}(t)_{i,p\neq\kappa}=0} \tag{2}$$

This architecture ensures that only one of the $k_i$ expanded parts is activated at each time step during sampling, thereby eliminating any additional computational overhead.

Figure 3 illustrates the operational mechanism of MoEDM. Specifically, we first discard mid-layers with the highest number of parameters. They usually hold more than 70% of the parameters. Subsequently, each remaining layer, $l_i$, will be transferred into a super-layer. Such a super-layer encompasses $k_i$ separate layers $(l_{i,0}, \ l_{i,1}, \ ..., \ l_{i,k_i-1})$ structured identically to $l_i$. For example, considering a super-layer comprising 2 independent separate layers, during the sampling process with a total of $T$ time steps, for $t_1$ in the first $\frac{T}{k_i}$ time steps, $l_{i,0}$ will be directly activated by the training-free gated selection mechanism in Equation 2, $\mathcal{G}(t_1)_{i,0} = 1$, to process the images. And $l_{i,1}$ will process the images for $t_2$ in the second $\frac{T}{k_i}$ time steps where $\mathcal{G}(t_2)_{i,1} = 1$, and so on. Similarly, during the fine-tuning process, only the images input corresponding to the time step of layer $l_{i,k}$ will be used to compute the gradient for that particular layer. While, the gradients computed using other images will be set to 0.

It is crucial to mention that the gating mechanism of MoEDM serves as a *training-free* approach. In diffusion models, the time step $t$ is always known, allowing for targeted activations based on $t$ without any concern about the degradation of dynamic gating mechanisms into static models. Additionally, by simply modifying the selection mechanism, we are able to set any expansion ratio for any layer at will, highlighting the system's flexibility compared to models like SDXL (Podell et al., 2023).

Although our initial tests indicate success, there remains room for further refinement, specifically in the automated training of an optimal layer expansion strategy. For example, we can assign larger amplification multiples for layers near both ends. This will form the cornerstone of our future research endeavors. For more details, please refer to the Appendix Section D.

**Distillation** Undoubtedly, data quality is a pivotal factor when fine-tuning MoEDM with a little dataset. Certain sub-classes within the ImageNet contain low-quality images, necessitating the inclusion of additional high-quality images from other sub-classes during fine-tuning. For example, a fraction (3/8) of training batch comes from from other high-quality sub-classes. However, many high-quality data categories usually remain off-limits to general users. To address this, we introduce a distillation technique in the training process, leveraging Latent Diffusion as a higher resolution base. Specifically, we use the full-scale Latent Diffusion model to generate sufficient task-specific images (e.g., 1,000 images), which then serve as a new training set for MoEDM. This leads to the redefinition of the optimization target, $\mathcal{L}'_d$, formulated as:

$$\mathcal{L}'_d = \| \ \mathcal{O}'(z_t, t, g) - \mathcal{O}(z_t, t, g) \ \|_2 \tag{3}$$

Here, $\mathcal{O}'$ and $\mathcal{O}$ represent the output from MoEDM and the full-size model, respectively. In the context of category label tasks, we employ the specific label as a query to gather the required images. For text-to-image tasks, we enlist the assistance of the Generative Pre-trained Transformer (GPT) (OpenAI, 2023). Specifically, we furnish GPT-3.5 with structured prompts designed to generate a diverse array of textual descriptions, facilitating the collection of images centered around a predefined theme. Importantly, we promise to ensure that the generated images are fully randomized and there will be no random seeds and text prompts that are the same as the sampling stage.

### 3.3 Summary

Our proposed method initially eliminates a substantial portion of parameters by pruning redundant layers, thereby markedly accelerating the sampling process for targeted applications. Following this, we integrate the mechanism of training-free dynamic routing to refine the fidelity of generated images, achieving this improvement without incurring extra computational burden.

## 4 Experiment

In this section, we report the experimental results of MoEDM across multiple tasks. These tasks encompass category-specific image generation, domain shift and text-to-image generation tasks. Our training experiments are conducted on 8 NVIDIA A100 GPUs. And the sampling is executed on a single NVIDIA A100 GPU (also run quickly on other low-memory GPUs, such as NVIDIA RTX 4090). For more details, including hyper-parameters, parameters count and memory usage, please refer to the Appendix Sections B and C.

### 4.1 Setup

**Models**  The MoEDM framework builds on existing work in both Guided Diffusion (Dhariwal & Nichol, 2021) and Latent Diffusion (Rombach et al., 2022), and follows the corresponding hyperparameters. Guided Diffusion serves as a foundational model for tasks involving category labels. To showcase the adaptability of MoEDM, we integrate it with Latent Diffusion, applying it to both category-label and text-to-image tasks. Furthermore, we employ DPM-Solver (Lu et al., 2022) and DDIM (Song et al., 2020) to facilitate accelerated sampling and underline the seamless compatibility of our approach with prevalent diffusion models.

**Baselines**  We compare our MoEDM with the corresponding full-size diffusion models. We have also integrated several classical experiments into our list of baseline models. These include training MoEDM from scratch, fully fine-tuning, BitFit (Zaken et al., 2021) (one of PEFT methods, Parameter Efficient Fine-Tuning) and partial blocks fine-tuning (one of PEFT methods, also known as "sparse fine-tuning," involves fine-tuning only a small subset of parameters, such as LoRA (Hu et al., 2021)). Due to constraints in paper presentation space, we only present the experimental results of these last 4 baselines on the most challenging and representative Domain Shift task.

**Evaluation Metrics**  We evaluate our method mainly on *FID* (Parmar et al., 2022) on 20,000 generated images with the ImageNet and FFHQ dataset. However, due to the constrained size of the Imagenet dataset (1,300 images per class), the computed *FID* results are not entirely precise. Therefore, we also report the *KID* (Salimans et al., 2016) results and offer additional visualizations to aid in evaluating the quality of images generated by MoEDM. Simultaneously, the *Runtime* also serve as a crucial evaluation metric. When calculating the runtime, we uniformly use a batch size of 4 for sampling.

### 4.2 Parameter Scoring

We identify parameters holding minimal significance for a specific task in Guided Diffusion $256 \times 256$ using Equation 1. We compare this result with the classical value-based method (Han et al., 2015) and gradient-based (Liu et al., 2021) method. And we also calculate the percentage from mid-layers of all discarded parameters. Experimental results suggest that mid-layers hold noticeably least overall significance, allowing us to implement our MoEDM by discarding mid-layers. For detailed experimental results, please refer to Table 4 and Figure 6 in the Appendix Section A.

### 4.3 Mixture of Expert Diffusion Models

### 4.3.1 Guided Diffusion

**Domain Shift and Baseline**  We initially report the performance of MoEDM in the most representative domain shift task, along with a series of experimental results comparing it to various baselines. Our baselines encompass training MoEDM from scratch, fully fine-tuning (acting as Dreambooth (Ruiz et al., 2022), a fully fine-tuning personalized method) and 2 PEFT methods: BitFit (Zaken et al., 2021) and partial blocks fine-tuning. For images not existing in the original training sets, our MoEDM can efficiently learn the image distribution in new data. We use Guided Diffusion at a resolution of $64 \times 64$ trained on ImageNet, and transfer it to FFHQ. We report these experimental results in Table 1. For the visualization of generated images, please refer to the Appendix Section F. Note that layers in these MoEDM models utilize an expand ratio of $2\times$. In this task, MoEDM achieves outstanding *FID* and *KID* scores with only a fraction of the training iterations, and it also doubles the sampling speed.

Table 1: *FID* (↓), *KID* (↓) and average time feedforward (↓) using MoEDM based on Guided Diffusion on the task of domain shift from ImageNet to FFHQ. Furthermore, we provide our ablation experiments and multiple baseline experiments in this challenging and representative task. Our baselines include fully fine-tuning, training MoEDM from scratch, BitFit and partially fine-tuning.

| Method | Ratio of Param to be Trained | *FID* ↓ | *KID* ↓ | Average Time FeedForward ↓ | Fine-tuning Iterations ↓ |
|---|---|---|---|---|---|
| MoEDM | 0.29 | **22.45** | **0.019** | **32.1 ms** | **1,100** |
| Fully Fine-tune (like DreamBooth) | 1.0 | 23.85 | 0.020 | 70.2 ms | 1,200 |
| MoEDM w/o Dynamic | 0.29 | 30.12 | 0.027 | **32.1 ms** | 1,500 |
| MoEDM from scratch | 0.29 | 22.70 | 0.020 | **32.1 ms** | 3,500 |
| BitFit (PEFT) | 0.006 | 36.69 | 0.032 | 70.2 ms | 10,000 |
| Partially Fine-tune (PEFT) | 0.29 | 23.75 | **0.019** | 70.2 ms | **1,100** |

In this challenging and representative task of Domain Shift, we report the results of MoEDM compared to baseline models, along with the results of ablation experiments. According to Table 1, training a MoEDM from scratch will not lead to a significant improvement in the model's performance, but it will incur substantial additional fine-tuning iterations; while methods like fully fine-tuning and PEFT can achieve good results, they fail to address the fundamental issue of the large number of parameters during the sampling stage, which limits the improvement in sampling speed.

In our ablation experiments, we ablate two components of MoEDM to show their contribution. On the one hand, as shown at the third line of Table 1, the model's performance is significantly below the acceptable standard demonstrated by the full-size models. On the other hand, to be undisputed, without discarding a significant portion of parameters, diffusion models cannot enhance the sampling speed anymore, and it might even escalate the training cost within the dynamic routing framework.

**Subsets of ImageNet** We first conduct experiments with MoEDM based on Guided Diffusion at various resolutions with subsets of ImageNet as specific tasks, where we randomly select subsets including artificialities, animals and plants. We quantitatively evaluate the performance of MoEDM using *FID* (↓), *KID* (↓) and Average Time Feedforward (↓), while also conducting manual observations of the visualizations generated by models. We report experimental results in Table2. In addition, we demonstrate the visualization images in Figure 8 in the Appendix Section F. Note that layers in these MoEDM models utilize an expand ratio of 2×.

Table 2: *FID* (↓), *KID* (↓) and average time feedforward (↓) using MoEDM based on Guided Diffusion on 3 random subsets of ImageNet, including artificialities, animals and plants. The symbols 1, 2, 3 and 4 respectively denote ImageNet's subsets of "School bus", "Cauliflower", "African elephant" and "Golden retriever". The symbol * denotes that we introduce images from other subsets in the training set as a way to illustrate the importance of the quality of training data.

| Resolution | Label | *FID* ↓ | | *KID* ↓ | | Average Time FeedForward ↓ | |
|---|---|---|---|---|---|---|---|
| | | Full-size | MoEDM | Full-size | MoEDM | Full-size | MoEDM |
| 64 × 64 | 779[1] | 44.80 | **38.16** | 0.029 | **0.022** | 70.2 ms | **32.1 ms** |
| 64 × 64 | 938[2] | 55.09 | **50.22** | 0.051 | **0.049** | 70.2 ms | **32.1 ms** |
| 64 × 64 | 386[3] | 67.57 | **62.53** | 0.046 | **0.041** | 70.2 ms | **32.1 ms** |
| 256 × 256 | 207[4] | **21.92** | 25.92 | **0.034** | 0.035 | 129.4 ms | **112.0 ms** |
| 256 × 256* | 207[4] | **21.92** | 22.21 | 0.034 | **0.033** | 129.4 ms | **112.0 ms** |

At a resolution of 64 × 64, MoEDM consistently outperforms full-size Guided Diffusion in specific tasks, showing improvements in both *FID* and *KID* scores. Furthermore, MoEDM also delivers a 2× sampling speed for each inference step. However, at a resolution of 256 × 256, while MoEDM achieves comparable

performance, the improvement in sampling speed is not as significant. In pixel space, parameters at both ends of the model handles significantly larger-sized images compared to mid-layers. This leads to a situation where those important layers with fewer parameters perform far more computation compared to mid-layers. While, Latent Diffusion transfers the diffusion process to the latent space, eliminating such issues.

### 4.3.2 Latent Diffusion

As outlined in Section 4.3.1, Latent Diffusion makes it possible for MoEDM to function effectively at high resolutions. At the same time, Latent Diffusion serves as a classical lightweight diffusion model, and experiments using MoEDM based on Latent Diffusion demonstrate its exceptional performance and compatibility. Furthermore, owing to the inclusion of the distillation approach, we will also include the baseline results of distillation in Table 7 in the Appendix Section D.

**Subsets of ImageNet**  We first conduct experiments with MoEDM based on Latent Diffusion at a resolutions of $256 \times 256$ with subsets of ImageNet as specific tasks. At the same time, we also compare the experimental results of whether distillation is introduced or not. We report these experimental results in Table 3. We also demonstrate the visualization of the images in Figure 10 in the Appendix Section F. Note that layers in these MoEDM models utilize an expand ratio of $2\times$.

Table 3: *FID* ($\downarrow$), *KID* ($\downarrow$) and average time feedforward ($\downarrow$) using MoEDM based on Latent Diffusion on subsets of ImageNet task and text-to-image task. The symbols 1, 2 and 3 respectively denote ImageNet's subsets of "Cheeseburger", "Head cabbage" and "Black swan". The symbol 3* denotes the experimental results of MoEDM trained without distillation, as a controlled experiment. The symbol 4 denotes various forms of prompts centered around a fixed character. Note that we use the same MoEDM model to generate images with different prompts.

| Guidance | *FID* $\downarrow$ | | *KID* $\downarrow$ | | Average Time Feedforward $\downarrow$ | |
| --- | --- | --- | --- | --- | --- | --- |
| | Full-size | MoEDM | Full-size | MoEDM | Full-size | MoEDM |
| Label 933[1] | 32.02 | **31.63** | 0.019 | **0.017** | 68.82 ms | **35.11 ms** |
| Label 936[2] | 63.62 | **60.85** | 0.033 | **0.031** | 68.82 ms | **35.11 ms** |
| Label 100[3] | 12.64 | **10.94** | 0.005 | **0.004** | 68.82 ms | **35.11 ms** |
| Label 100[3*] | **12.64** | 67.90 | **0.005** | 0.052 | 68.82 ms | **35.11 ms** |
| Text Prompt [4] | - | - | - | - | 73.36 ms | **37.21 ms** |

Compared to MoEDM w/o distillation, the integration of distillation has led to a notable enhancement in the performance of MoEDM. Compared to full-size models, whether it's *FID* or *KID* scores in specific tasks, MoEDM outperformes full-size models and provides a $2\times$ sampling speed.

**Text-to-image**  To further demonstrate the value of MoEDM in real-world applications, we conduct experiments involving text-to-image generation. The primary distinction between label-to-image and text-to-image tasks lies in the ability of text-to-image tasks to amalgamate different concepts within a single image. In MoEDM, a specific task is defined as a fixed concept in different environment concepts. Given that the pre-trained model has acquired the skill of merging different concepts, we can directly fine-tune MoEDM with images featuring diverse concepts, rather than using completely fixed prompts. Given the constraints of *FID* and Clipscore in text-to-image tasks, we propose to evaluate the quality of image generation in this task by human eyes. We report the enhancement of the sampling speed in Table 3. In Figure 4 and 5, we present the visualization of generated images, and we also present more visualization results in the Appendix Section F. They all yield positive results, providing robust support for our MoEDM.

**Uneven Expansion**  As mentioned in Section 3.2, utilizing an identical ratio of expansion for all remaining layers might not be the optimal strategy. Training such a strategy is not easy, so we provisionally try manual specify expansion ratio for different layers. We report detailed experimental results in Table 7 in the Appendix Section D, please refer it for more details.

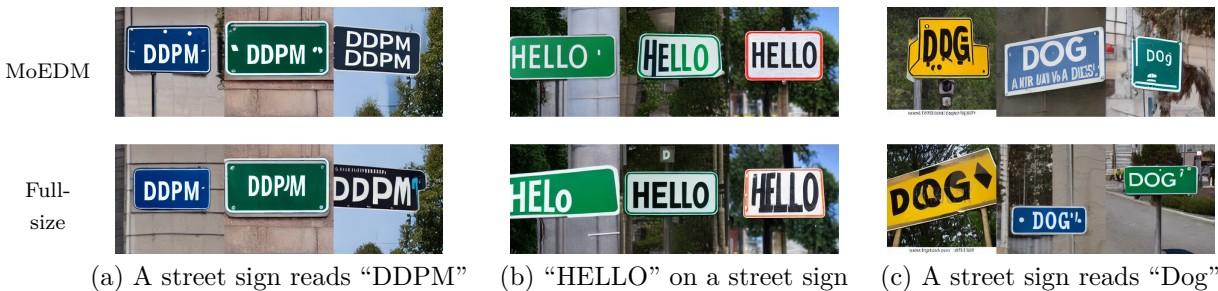

(a) A street sign reads "DDPM"     (b) "HELLO" on a street sign     (c) A street sign reads "Dog"

Figure 4: Visualization of images generated by the same MoEDM based on Latent Diffusion $256 \times 256$ on the task of text-to-image. We use common words as prompts to fine-tune MoEDM, and make sure the presentation uses words that do not appear in the training data at all.

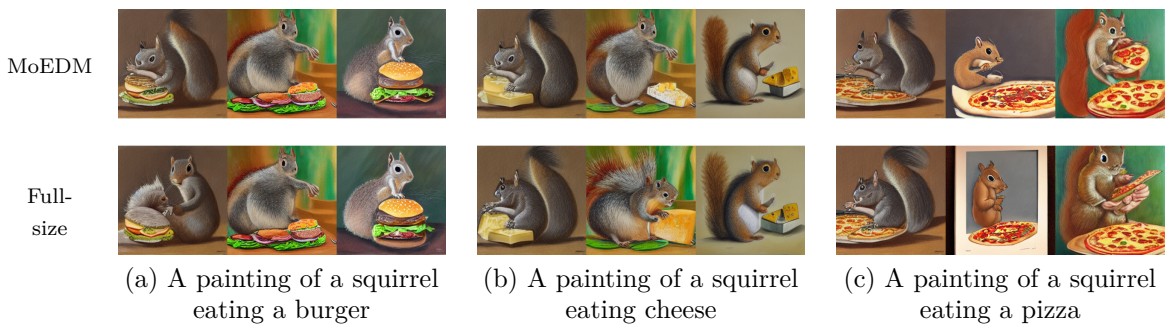

(a) A painting of a squirrel eating a burger     (b) A painting of a squirrel eating cheese     (c) A painting of a squirrel eating a pizza

Figure 5: Visualization of paintings of a squirrel eating various fast food generated by the same MoEDM based on Latent Diffusion $256 \times 256$ on the task of text-to-image. We train MoEDM using images of "a painting of a squirrel is eating" and various common fast food instead of training with images of fixed prompts, "a painting of a squirrel eating [food]".

## 5   Discussion

In conclusion, we have proposed a novel method (MoEDM) for lightening and customizing personalized diffusion models, which is a new effective perspective for addressing deployment challenges of diffusion models. Our MoEDM makes a notable reduction in parameters and enhancement in sampling speed for specific tasks while preserving the performance of image generation. Moreover, MoEDM can be seamlessly integrated with various existing user-friendly diffusion models, e.g., DPM-Solver (Lu et al., 2022), DDIM (Song et al., 2020) and Latent Diffusion (Rombach et al., 2022), demonstrating its excellent scalability. While, the strict one-hot gated strategy and the uniform expansion strategy in fine-tuning imposes limitations on the efficiency of the fine-tuning procedure. Addressing these challenges could potentially involve employing techniques such as parameter or gradient sharing, alongside training an expansion strategy. These aspects will constitute the primary focus of our forthcoming research endeavors.

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

# A   Score of Layers

In this section, we will present more detailed relationship between the significance of parameters in Guided Diffusion $256 \times 256$ and the quality of the generated images. We identify parameters holding minimal significance for a specific task using Equation 1, and subsequently set the designated number of parameters to zero (equivalent to discarding them). We compare this result with the classical value-based method (Han et al., 2015) and gradient-based (Liu et al., 2021) method. We also calculate the percentage from mid-layers of all discarded parameters. We report the experimental results for parameter scoring in Table 4. Note that the results in Table 4 are obtained directly through sampling after parameter discarding, without any additional training. When the discard ratio is as low as 0.1, the classical approach based on feedforward models becomes ineffective. While, our approach can, to a certain extent, maintain the model's performance. In comparison with the other 2 methods, most of parameters discarded by our approach originate from mid-layers.

Table 4: *FID* (↓) of different scoring methods at a low discard ratio in Guided Diffusion $256 \times 256$. Here we discard insignificant parameters provided by 3 scoring methods and directly calculate the *FID* of the generated images without any additional training. We also count the proportion of insignificant parameters from mid-layers to determine whether these layers held the least significance.

| Method | Discard Ratio | *FID* ↓ | Percentage from Mid-layers |
|---|---|---|---|
| Full-size | - | 21.92 | - |
| Value-based | 0.10 | 459.35 | 0.49 |
| Grad-based | 0.10 | 412.31 | 0.69 |
| Ours | 0.10 | **28.27** | 0.94 |
| Ours | 0.20 | 89.37 | 0.90 |

In addition, we present a more intuitive visualization of the results. In Figure 6, the height of each histogram show the proportion of parameters from the current layer in the total discarded parameters. From this standpoint, the model's performance can be optimally preserved when the most of the discarded parameters are drawn from mid-layers.

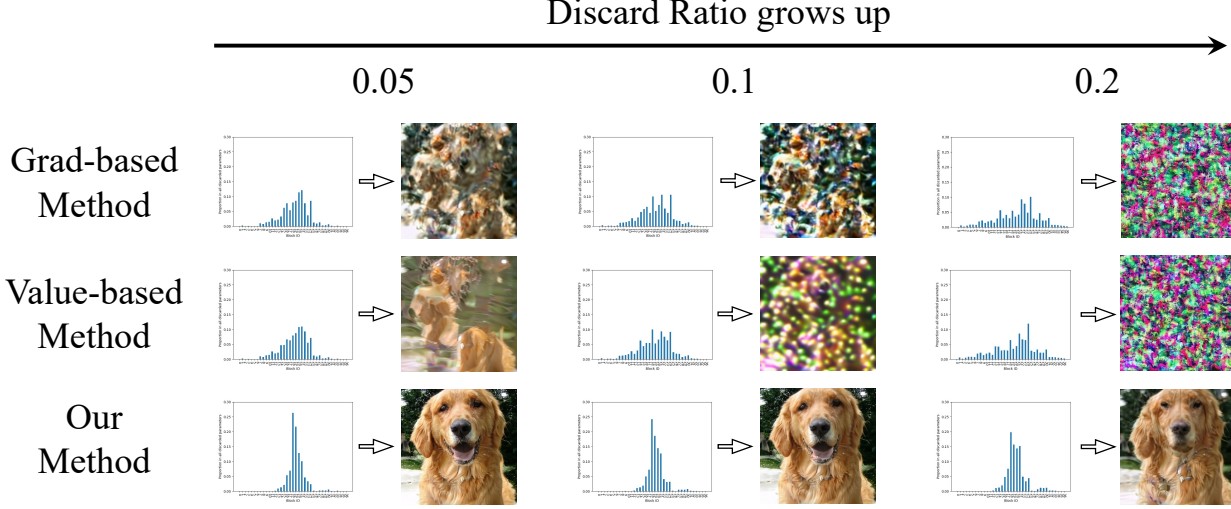

Figure 6: The relationship between the significance of parameters in Guided Diffusion $256 \times 256$ and the quality of the generated images. The higher histogram shows the more discarded parameters come from the current block. When the discarded parameters are concentrated in middle layers, the model tends to generate significantly higher-quality images.

## B  Hyper-Parameters

Here we report the key hyper-parameters of MoEDM in Table 5. We follow the hyper-parameters used in Guided Diffusion and Latent Diffusion. Discarding a substantial number of parameters and focusing on specialized tasks, the fine-tuning process of MoEDM requires only very few iterations. Simultaneously, we have also made efforts to reduce the fine-tuning duration. First we fine-tune the pruned model using noised images from all time steps, followed by fine-tuning the corresponding modules in MoEDM using noised images from different time steps. This approach results in a one-third reduction in the originally planned fine-tuning time.

Table 5: The training hyper-parameters of full-size diffusion models and MoEDM.

| Model | Image Size | Batch Size | Learning Rate | Training Iteration |
|---|---|---|---|---|
| Guide Diffusion | $64 \times 64$ | 2,048 | 3e-4 | 540,000 |
| Latent Diffusion | $256 \times 256$ | 1,200 | 1e-4 | 178,000 |
| Latent Diffusion (text-to-image) | $256 \times 256$ | 1,200 | 1e-4 | 390,000 |
| MoEDM (Guided) | $64 \times 64$ | 2,048 | 3e-4 | 1,800 |
| MoEDM (Latent) | $256 \times 256$ | 1,200 | 1e-4 | 2,500 |
| MoEDM (Latent, text-to-image) | $256 \times 256$ | 1,200 | 1e-4 | 3,000 |

## C  Computational Requirements

Runtime memory requirement is essential to a modern machine learning application. After discarding unimportant parameters in diffusion models and expanding the remaining layers, we emphasize that these operations do not result in an increase in memory usage. Instead, there is some relief in terms of memory usage. We report the discard ratio ($\uparrow$) and the memory usage ($\downarrow$) of MoEDM in Table 6. Note that we use a batch size of 4 when reporting the memory usage.

Table 6: The discard ratio and the memory usage of full-size diffusion models and MoEDM.

| Model | Image Size | Discard Ratio $\uparrow$ | Memory Usage$\downarrow$ |
|---|---|---|---|
| Guided Diffusion | $64 \times 64$ | Full size | 3965M |
| Latent Diffusion | $256 \times 256$ | Full size | 5603M |
| Latent Diffusion (text-to-image) | $256 \times 256$ | Full size | 9167M |
| MoEDM (Guided) | $64 \times 64$ | 0.71 | 3689M |
| MoEDM (Latent) | $256 \times 256$ | 0.77 | 4094M |
| MoEDM (Latent, text-to-image) | $256 \times 256$ | 0.84 | 7261M |

## D  Uneven Expansion

As mentioned in Section 3.2, utilizing an identical ratio of expansion for all remaining layers might not be the optimal strategy. Training such a strategy is not easy, so we provisionally try manual specify expansion ratio for different layers. We allocate a $2\times$ expansion ratio to the 6 layers positioned closer to the input-output stages within the remaining 12 layers of Latent Diffusion. We allocate a $2\times$ expansion ratio to the 6 layers

Table 7: *FID* (↓) and *KID* (↓) when using uniform expansion and uneven expansion in MoEDM.

| Expansion Strategy | *FID* ↓ | *KID* ↓ |
|---|---|---|
| Uniform 2× | 10.94 | 0.004 |
| Uneven | 11.51 | 0.005 |
| w/o Dynamic | 16.82 | 0.008 |

positioned closer to the input-output stages within the remaining 12 layers of Latent Diffusion. Simultaneously, the other 6 layers will not be expanded at all. We compare it with experiments involving uniform expansion with a 2× ratio and a baseline w/o dynamic, respectively. We report the experimental results in Table 7. The uneven expansion maintains model's performance and effectively decreases the quantity of parameters requiring training and storage. This result has reinforced our determination to optimizing expansion strategies in the future.

## E  Discard Ratio

Here we present images generated by MoEDM with different discard ratios in Figure 7. All of these models have undergone sufficient training. Even MoEDM with the highest discard ratio (a) ensures faithful generation of the specified contents, albeit with some minor details at the edges possibly missing.

Feedforward Time: 37.21ms     Feedforward Time: 41.13ms     Feedforward Time: 49.85ms

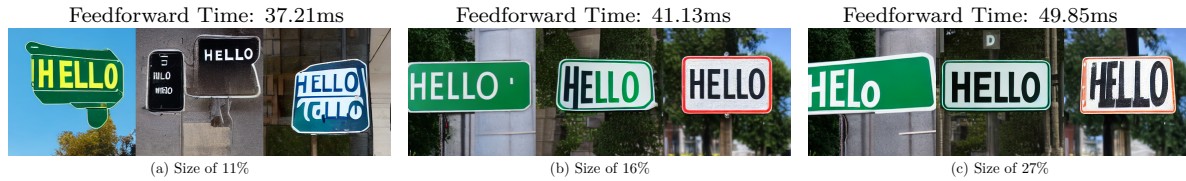

(a) Size of 11%     (b) Size of 16%     (c) Size of 27%

Figure 7: Visualization of images generated by MoEDM with different discard ratios.

## F  Image Visualization

Here we demonstrate the visualizations of images generated by MoEDM in Figure 8, 9, 10, 11, 12. Additionally, In addition, we also demonstrate lots of images generated by MoEDM in Figures 13, 14, 15 and 16.

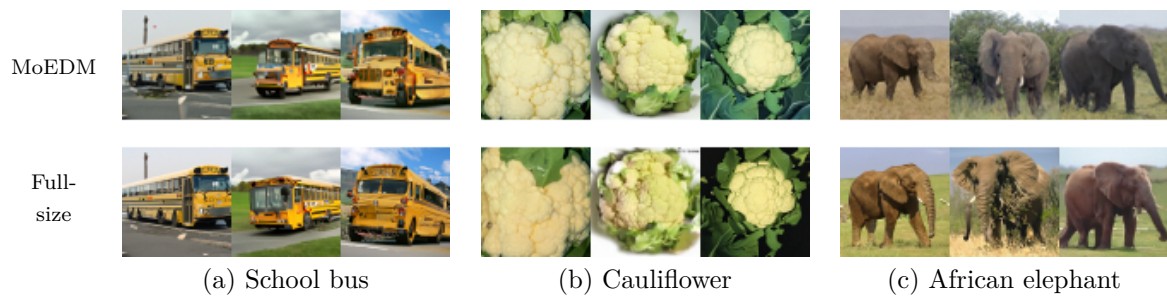

MoEDM

Full-size

(a) School bus     (b) Cauliflower     (c) African elephant

Figure 8: Visualization of images generated by MoEDM based on Guided Diffusion 64 × 64 on 3 random subsets of ImageNet, including artificialities, animals and plants. The first row presents images generated by MoEDM, and the second presents images generated by full-size Guided Diffusion with a same random seed.

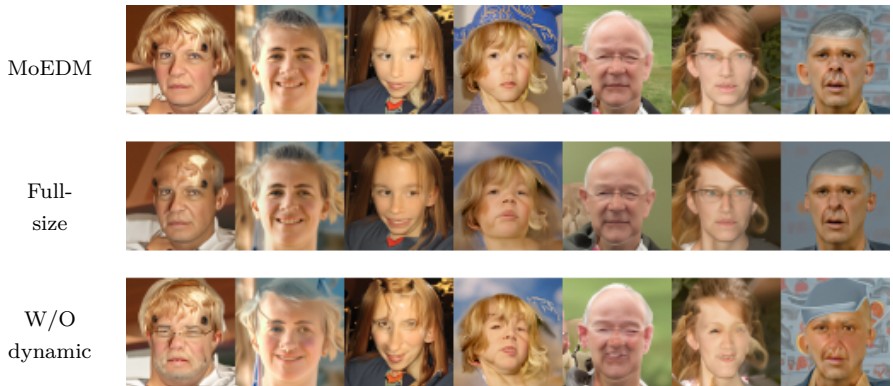

Figure 9: Visualization of images generated by MoEDM based on Guided Diffusion $64 \times 64$ on the specifc task of domain shift from ImageNet to FFHQ. The first row presents images generated by MoEDM, and the second row presents images generated by full-size Guided Diffusion with a same random seed.

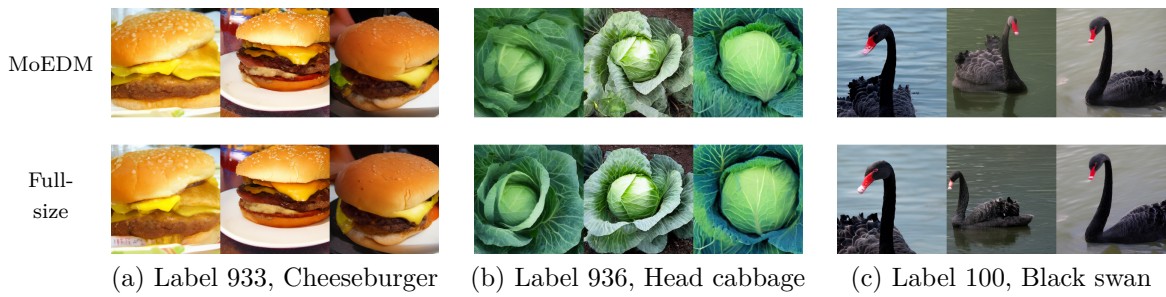

(a) Label 933, Cheeseburger    (b) Label 936, Head cabbage    (c) Label 100, Black swan

Figure 10: Visualization of images generated by MoEDM based on Latent Diffusion $256 \times 256$ on 3 random subsets of ImageNet, including artificialities, animals and plants. The first row presents images generated by MoEDM, and the second presents images generated by full-size Latent Diffusion with a same random seed.

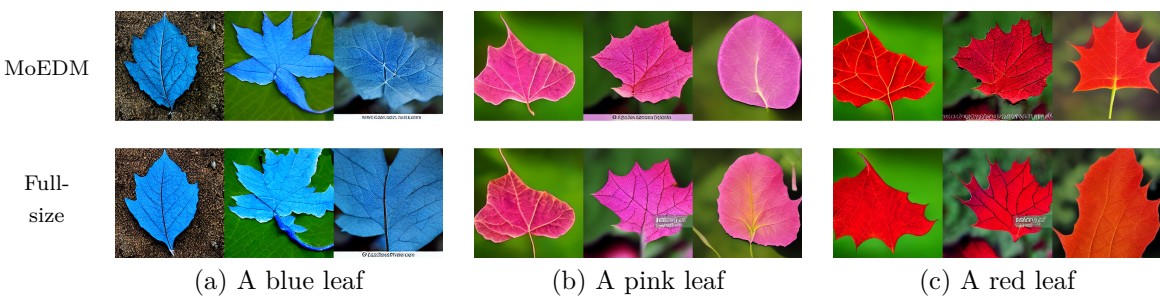

(a) A blue leaf                (b) A pink leaf                (c) A red leaf

Figure 11: Visualization of images of leaves in different colors generated by the same MoEDM based on Latent Diffusion $256 \times 256$ on the task of text-to-image. Here, we train MoEDM using images of "a leaf", "red color", "blue color" and various common colors instead of training with images of fixed prompts, "a [color] leaf".

MoEDM

Full-size

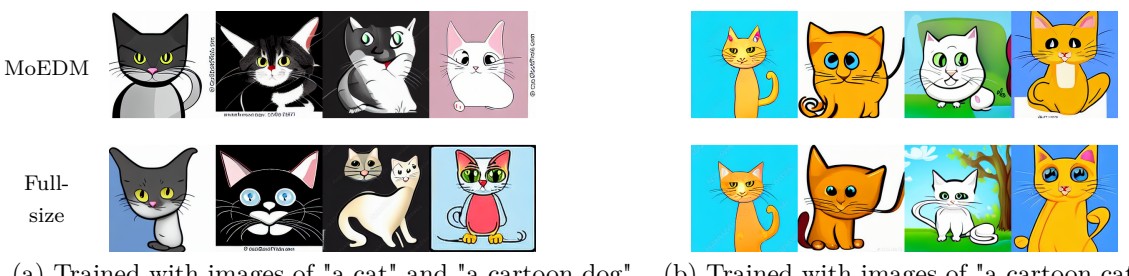

(a) Trained with images of "a cat" and "a cartoon dog"  (b) Trained with images of "a cartoon cat"

Figure 12: Visualization of images of a cartoon cat generated by MoEDM based on Latent Diffusion $256 \times 256$ on the task of text-to-image. Here, we train MoEDM using images of "a cat" and "a cartoon dog". MoEDM acquires the style of cartoon from cartoon dogs and accurately applies it when tasked with creating "a cartoon cat".

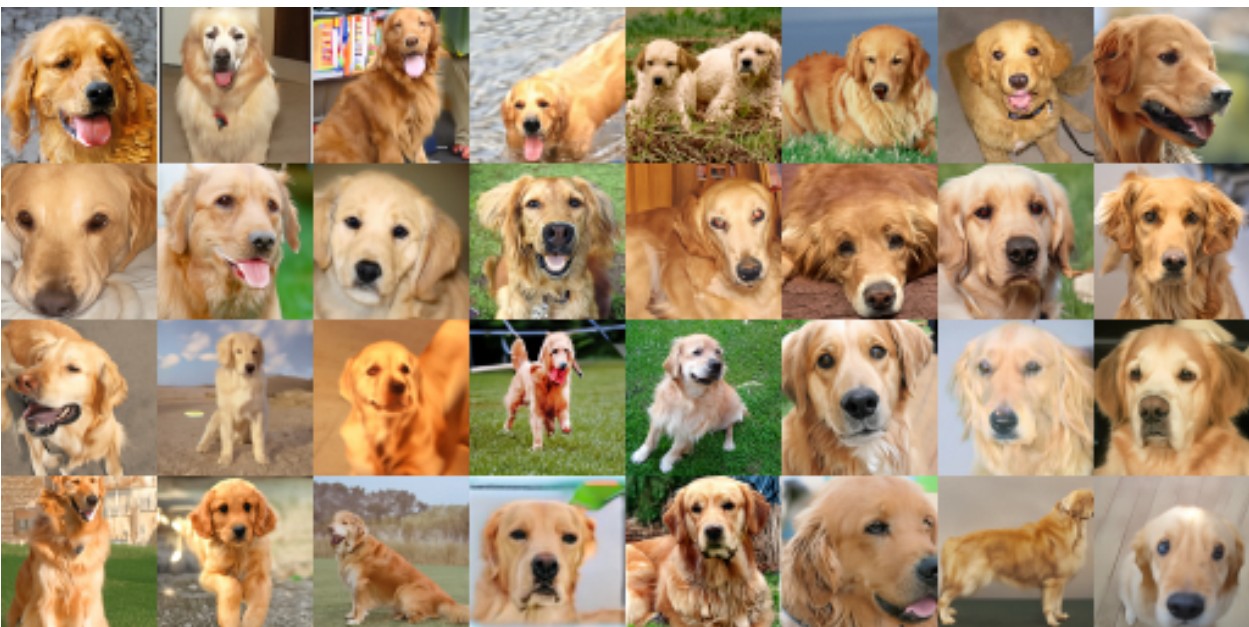

Figure 13: Visualization of images generated by MoEDM based on Guided Diffusion $64 \times 64$, guided by the label of "Golden Retriever".

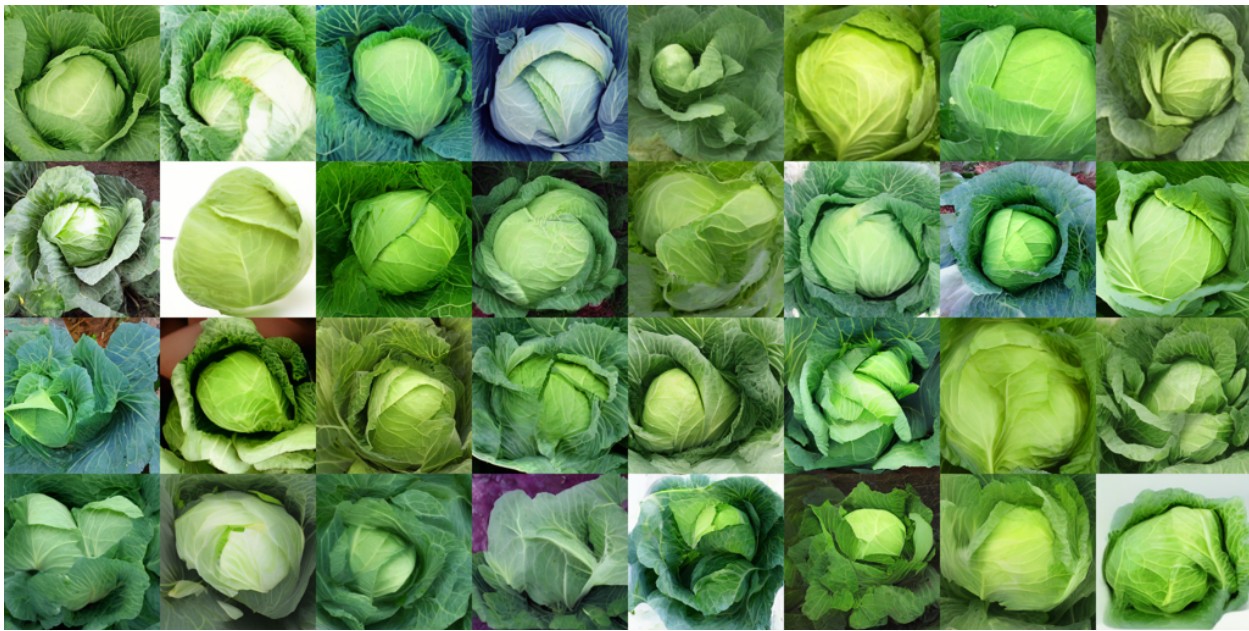

Figure 14: Visualization of images generated by MoEDM based on Latent Diffusion $256 \times 256$, guided by the label of "Cabbage".

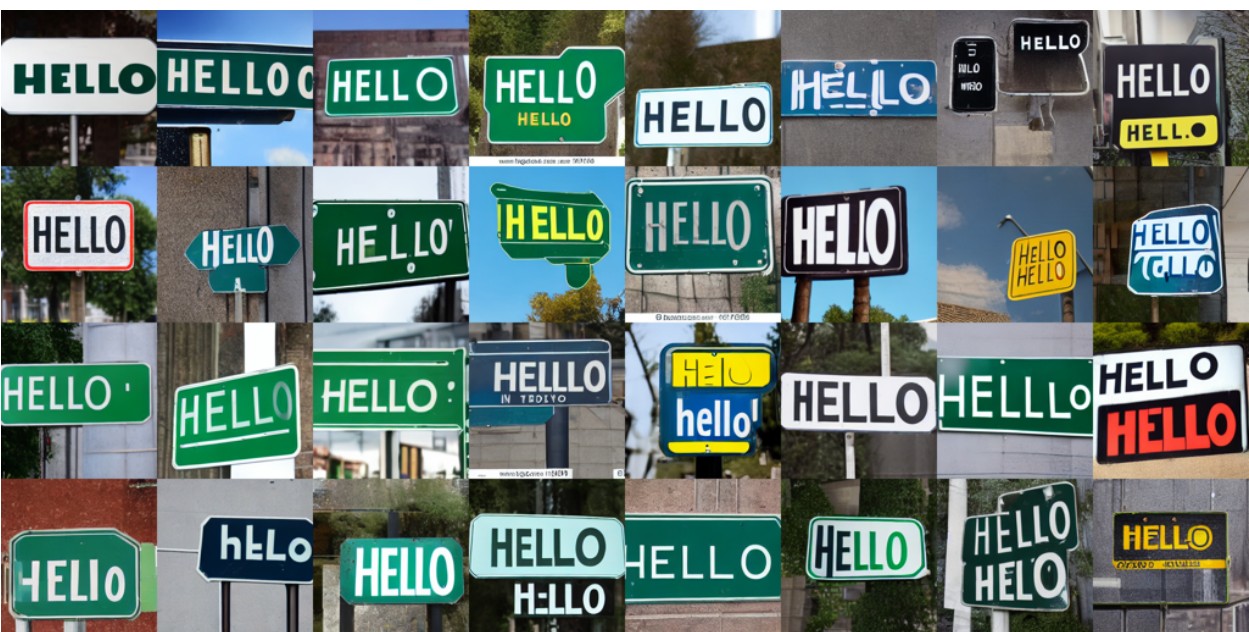

Figure 15: Visualization of images generated by MoEDM based on Latent Diffusion $256 \times 256$, guided by the text prompt "street sign reads Hello".

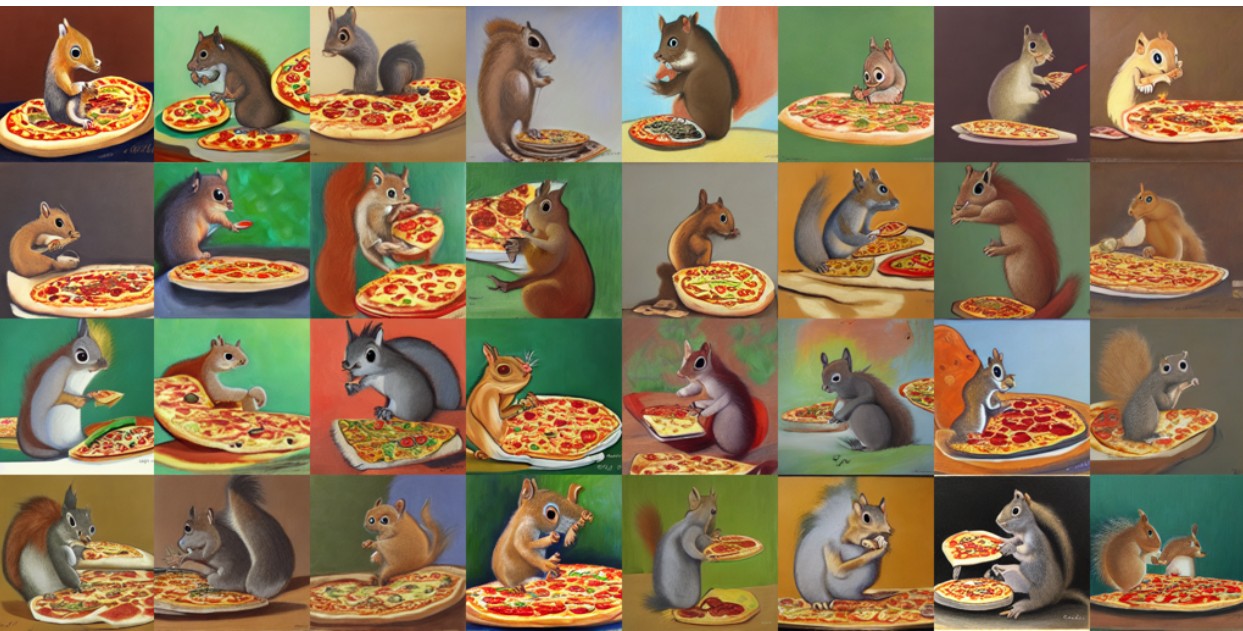

Figure 16: Visualization of images generated by MoEDM based on Latent Diffusion $256 \times 256$, guided by the text prompt "a painting of a squirrel eating a pizza".

