# OpenReview forum: "Experts on Demand: Dynamic Routing for Personalized Diffusion Models"
_TMLR — Rejected by TMLR_

### Review · Reviewer_YeiB · 2024-02-14

**Summary Of Contributions:**

The submission proposes a dynamic routing approach (MoEDM) for fast and lightweight image generation with diffusion models. The proposed method first discards unimportant parameters in the middle layers of the network, and subsequently finetunes the reduced model with a fixed, training-free gating mechanism that sequentially activates the remaining layers during generation.

Performance is compared with the full-size original model on class-conditional image generation, text-to-image generation, and style transfer. Results show that MoEDM retains image generation quality while halving sampling time.

**Audience:**

Yes

**Claims And Evidence:**

Yes

**Requested Changes:**

**Questions on Eq. (1)**

* Notation: are $p_\theta$ and $p_{\theta'_c}$ probability distributions or images? If the former, then the $\ell_1$-distance between two distributions is related to the TV distance. If the latter, than it is the $\ell_1$-norm of a vector. This approach seems inspired by Eq. (3) and (4) in Liu et al., 2021. However, those equations are defined on activation vectors. So it is clear in their context what they are referring to. Could the authors clarify what objects are treated in Eq. (1)?
* Under either option above, Eq. (1) defines some notion of contribution of an individual channel $c$ to the generation process (either in terms of TV distance or norm). However, this quantity on its own does not define an optimization process to select the subset of channels to drop from the original diffusion model. My interpretation was that this notion of importance is used to select a subset of channels to remove such that the norm is minimized. There also seems to be the need of a constraint defined in terms of the "discard ratio", i.e. how many channels to drop.
* Once this optimization problem is defined, some relaxation seems to be needed to solve it efficiently, because selecting a subset of channels is generally NP-hard. Am I missing something here?
* Finally, I am confused on how this objective is estimated in practice. Going back to Liu et al., there, it seems that the objective function is defined over draws of noise. That is fine because GANs are deterministic functions of their input. Diffusion models, on the other hand, are stochastic, so the initial draw of noise may generate different samples. Is the objective computed over a set of hold-out images whose likelihood is estimated by solving the probability flow ODE associated with the original and lightweight diffusion models?

---

**Minor comments**

* In the introduction, it is mentioned that "the instability of cloud servers" is a practical limitation. This is mostly out of curiosity: in the authors' experience, what instabilities have they faced that limited the practicality of diffusion models?
* Extra blank space in "mid-layers" on Page 4.
* It could be useful to briefly mention a few examples of what "guidance" refers to in the paragraph before Eq. (1).
* In the paragraph after Eq. (1), it is mentioned "regardless of the discard ratio". Intuitively, I think this means what percentage of layers one is willing to let go from the original diffusion model. However, this is not formally defined in the text, and Eq. (1) is presented not as a constrained optimization problem, should this be included?
* Sec. 3.2, Discard Layers paragraph: "each element is multiplicatively combined with the output from the corresponding ...". I interpreted this as a "layer-dropout". Do the authors agree with this interpretation? It could be useful to explain this with words as well.
* Sec. 3.2, Discard Layers paragraph: in the last sub-paragraph, it is mentioned that assigning "larger amplification multiples" may be a better strategy for "layer expansion". I am not sure I understand what these terms mean. Could the authors include an explanation of these terms?
* Sec. 4.3.1, Domain Shift and Baseline paragraph: typos in "2 PEFT methods ... of PEFT and ... of PEFT"? I am not sure I understand this sentence. Also, what is "partial blocks fine-tuning"? Is this a specific fine-tuning technique? If so, maybe a reference should be included?
* Table 1: "MoEDM w/o Dynamic" and "MoEDM from scratch" are unclear. In practice, what ablations are performed?
* Figure 4: missing label of which samples come from MoEDM or the full-size model.
* General question on example figures: I was surprised how similar the images from MoEDM and the full-size model were. Am I missing something here? I was expecting these figures to show a broad range of diverse images sampled from each model. Given only the prompts, I was surprised how the presented images almost show a one to one correspondence in their contents. Were these images cherry-picked to show this correspondence? Could the authors include more images in an Appendix, as it is usually done with papers that present new sampling strategies? I understand Appendix F shows images. Still, these images are very aligned across models.

**Strengths And Weaknesses:**

Strengths:
* The paper is well written and clear in its presentation
* The topic of personalized diffusion models is relevant
* Approach is novel and intriguing

Weaknesses:
* MoEDM still requires a (minimal) amount of finetuning of the lightweight network
* Limited discussion on how to solve problem in Eq. (1)

Specifically, I have a few questions regarding the optimization problem in Eq. (1). I will expand on these questions and I am looking forward to discussing with the authors!

---

> ### Author Response · Authors · 2024-02-16
> **Reply to Reviewer YeiB's Review**
>
> Thank you very much for the thoughtful and detailed review. We reply point-by-point here, to begin the discussion. Due to the word limit of a single text box, we will write our reply in several text boxes. And we will submit a revised version of our paper soon.
> 1. **"Need for Fine-tuning"**: Certainly, MoEDM does require some degree of fine-tuning, primarily because over 70% of the pre-trained parameters have been removed. However, as illustrated in Table 5 of the appendix in our paper, fine-tuning typically demands significantly fewer iterations. With the utilization of 8 Nvidia A100 GPUs, the process can be completed in less than 3 hours, which is reasonable for scenarios with long term, specific requirements, such as a pet store.
> **Simultaneously, we have also made efforts to reduce the fine-tuning duration.** First we fine-tune the pruned model using noised images from all time steps, followed by fine-tuning the corresponding modules in MoEDM using noised images from different time steps. This approach results in a one-third reduction in the originally planned fine-tuning time. Of course, we will also supplement this part in the updated paper.
>
> 2. About "Eq. (1)":
>     - **"Notation"**: We apologize for the confusion caused by our notation. In fact, what we intended to convey with $p_\theta$ and $p_{\theta'c}$ corresponds to the latter, as you mentioned. This implies images obtained through a complete round of diffusion sampling using the well-trained model and images obtained by sampling after removing channel $c$. We will revise the notation to $I_\theta$ and $I_{\theta'_c}$ in the updated paper.
>
>     - **"Discard Ratio": Your interpretation is correct. The discard ratio mentioned here is automatically determined based on the number of layers to be removed, in Section 3.2, specifically in the "Discard Layers" paragraph.** To elaborate, the purpose of Eq. (1) is to identify the parameters (channels) in the model that have the least impact on the results. After obtaining the importance of each channel, we initially attempted precise channel deletion. However, due to the presence of the Group Norm layer, aligning the number of channels in the model posed a significant challenge. Upon re-evaluating the importance of channels derived from Eq. (1), we observed that the majority of unimportant channels were concentrated in the middle layers. Consequently, we suggest directly removing these mid-layers, significantly enhancing the sampling speed. The discard ratio of parameters is automatically determined by the number of layers removed, as the trained model structure is deterministic. In this paper, we generally maintain the discard ratio between 70% and 80%. A detailed explanation is provided in the first paragraph of Section 3.2, supported by tables 4 and 6 in the appendix, along with Figure 6.
>     - **"NP Hard": Completing Eq. (1) is indeed a time-consuming task as it involves traversing the channels in the model. However, it is not necessary to perform it for every specific task, but it only needs to be performed once.** We conducted experiments using 5 random seeds to validate Eq. (1) across tasks under the unconditional model and all the condition for each specific task as presented in the paper. The conclusion drawn from over 50 experiments is consistent, supported by included code in the supplementary materials. At least 90% of the deleted parameters were found to originate from mid-layers. This operation is no longer required in the subsequent implementation of MoEDM and fine-tuning processes.
>
>         **We also attempted to accelerate this process by traversing channels in groups of 10.** The results indicated that 70% of the parameters to be deleted originated from mid-layers. However, when setting the same proportion of unimportant parameters to zero, the sampled image quality was noticeably inferior compared to the previous approach.
>
>     - **"Randomness"**: We did consider the stochastic nature of the diffusion model. In each experiment, we fixed the random seed and conducted the experiment five times with randomly selected different seeds. We used the same number of images as Liu et al., which is 100 images. Despite the time-consuming nature of this experiment, it yielded valuable insights, confirming that almost all deleted parameters come from mid-layers.
>
> We will give the reply to Minor Comments in the next text box.

---

> ### Author Response · Authors · 2024-02-16
> **Reply to Reviewer YeiB's Review (Minor Comments)**
>
> 3. Reply to Minor Comments:
>     - **"The Instability of Cloud Servers"**: We can give an example, for example, in November 2023, the famous OpenAI company experienced a server anomaly for several weeks, which made it very difficult to obtain the Dalle service it provided.
>     - **"Extra Blank"**: Thank you for pointing out this typo. We will rectify the extra blank space in "mid-layers" on Page 4 in our updated paper.
>     - **"Guidance"**: Thank you for pointing out the lack of explanation for the "guidance" in Eq. (1). We will provide the necessary clarification in our updated paper. It represents various forms of guidance, such as category labels or text prompts.
>     - **Thank you for pointing out the lack of explanation for "regardless of the discard ratio".** We will provide the necessary clarification in our updated paper. Specifically, after obtaining the importance of each channel, we proceed to rank their significance. Subsequently, regardless of the discard ratio $r_d$%, we observed that nearly all the top $r_d$% parameters to be deleted are from mid-layers. In other words, the parameters in mid-layers are almost entirely of secondary importance.
>
>       **What's more, we consider expressing it in a constrained form unnecessary since g is merely a parameter.** Moreover, the experiments introduced in point 3 of the second response above have demonstrated that the values of $g$ do not impact the distribution of scores. Anyway, we will provide more detailed explanations in our updated paper.
>
>     - **"Layer-dropout": Yes, we completely agree with your explanation.** We will provide the necessary explanation in word in our updated paper. When an element in the gated vector is set to 0, the output of the layer multiplied by it becomes 0, signifying the removal of that layer.
>
>       **It is important to note that** this paragraph validates the layer-discard strategy, whereas in the implementation of MoEDM, layers are directly removed rather than simulated by zeroing. We will provide a clearer explanation in our updated paper.
>
>     - **"Larger amplification multiples": We are happy to provide you with detailed explanations and add them to our updated paper.** Specifically, we extend each remaining layer in the model into a super layer, essentially replicating it $n$ times, and during sampling, different parts are activated to enhance speed and improve model performance. A higher $n$ improves performance but introduces a greater fine-tuning burden. Based on the conclusions in Section 3.1 "Parameter Scoring," layers closer to the edges are more crucial. Therefore, intuitively, among the remaining layers, we can assign a larger amplification multiple to the layers near the edges and reduce the amplification multiple for layers in the middle. This approach aims to further improve model performance without increasing the fine-tuning cost.
>
>       For example, suppose we currently have 18 layers, numbered 1-18. Layers 5-14 are permanently removed, and the remaining layers are uniformly extended by a factor of 2. The desired strategy is to further extend layers 1, 2, 17, and 18 by a factor of 4, while layers 3, 4, 15, and 16 remain unextended. Table 7 in the appendix provides supporting details, and this will be pursued as one of our future research directions.
>
>     - **"PEFT": Two variants of PEFT are employed in this study.** We will address the typos mentioned in your feedback in our updated paper. **Additionally, "partial blocks fine-tuning,"** also known as "sparse fine-tuning," involves fine-tuning only a small subset of parameters, such as LoRA[*1]. The choice of fine-tuning parameters is contingent upon the specific requirements of individual tasks. In the baseline experiments conducted in our paper, we exclusively fine-tuned the layers located on both ends of the remaining layers, constituting approximately half of the remaining layers.
>
>
> The remaining replies to Minor Comments will be given in the next text box.
>
> [*1] Hu, Edward J., et al. "Lora: Low-rank adaptation of large language models." arXiv preprint arXiv:2106.09685 (2021).

---

> ### Author Response · Authors · 2024-02-16
> **Reply to Reviewer YeiB's Review (Minor Comments)**
>
> 3. Reply to Minor Comments (following the previous text box):
>     - **"Ablation experiments"**: Here, "MoEDM w/o Dynamic" means we only remove mid-layers and not build dynamic routings; "MoEDM from scratch" means we train a MoEDM model from random initialization. We will provide the necessary clarification in our updated paper.
>
>       The ablation experiments in this paper primarily consist of "MoEDM w/o Dynamic". Another ablation, "constructing dynamic routing using the full-size model," is equivalent to fine-tuning the full-size model and does not contribute to any improvement in sampling speed fundamentally.
>
>     - **"Figure 4"**: Thank you for pointing out the issue in Figure 4. We will rectify this issue in the revised version of our paper.
>
>     - **"Aligned images"**: All images presented in the paper consist of at least two rows, where the images in the upper and lower rows correspond one-to-one. This arrangement is due to obtaining the upper and lower images using MoEDM and the full-size model with the same random seed. This setup facilitates a better comparison of the quality of images generated by the two models. It is observed that there are some differences in color and content between the upper and lower rows due to the use of different models with the same random seed. However, the overall quality of the images remains consistent, providing evidence for the effectiveness of MoEDM. Furthermore, we are open to showcasing a greater number of images in the updated version of the paper.

---

### Review · Reviewer_FQW7 · 2024-02-19

**Summary Of Contributions:**

The paper introduces Mixture of Expert Diffusion Model (MoEDM), an efficient strategy for fine-tuning diffusion models. Instead of training with the full parameters of the model (full fine-tuning), MoEDM first prunes unimportant parameters (or channels) of the model at each time step and additionally perform distillation to obtain computationally and memory efficient yet capable diffusion models. Empirically, the authors show that MoEDM achieves competitive results with full-finetuning and other parameter-efficient fine-tuning baselines on several tasks, but with more efficient training and sampling.

**Audience:**

Yes

**Broader Impact Concerns:**

I do not believe the submission requires adding a broader impact statement.

**Claims And Evidence:**

Yes

**Requested Changes:**

- As mentioned above, it would be greatly helpful for the readers if the writing of the paper is improved. At the moment, the limited writing makes the paper difficult to understand.
- (Minor) It would be helpful if the authors provided a direct link to the Appendix (e.g., “... refer to the Appendix A” rather than “... “refer to the Appendix”).
- I would appreciate if the authors describe the limitations of MoEDM in the main text (e.g., showing failure cases if authors found any).

**Strengths And Weaknesses:**

Strengths:

- The subject of this paper aligns with the interests of TMLR's audience. While the paper is overall well-motivated, the writing can be significantly improved, as elaborated on below in the weaknesses section.
- The claims mentioned in the paper are justified and convincing. The proposed approach (MoEDM) intuitively makes sense as an approach to make diffusion models more efficient in terms of computational overheads and memory.
- The authors provided the code to reproduce some experiments in the paper. Furthermore, the hyperparameter configurations and how they were selected are reported in the Appendix (although I believe that the authors should conduct additional grid searches to find the optimal hyperparameters, such as the learning rate for each technique, instead of using a fixed constant value for all techniques).

Weaknesses

- The writing can be significantly improved by making statements and descriptions concise. For example, sentences such as “While, do users truly hunger for such an all-encompassing arsenal”, “micro-managing model parameters, …”, and “Navigating the path to this optimal blend of efficiency and functionality is complex, fraught with hurdles including the inherent time-dependent complexities …” sound awkward, and these sentences appear multiple times throughout the paper. In many cases, these make it difficult to understand what authors are trying to argue. At the moment, the writing is the biggest weakness of the submitted manuscript.
- Similarly, the experiment section in the paper is not well organized, and the above writing issues make the setup and task difficult to understand fully. While I am not very familiar with the diffusion literature, tasks that authors investigate (e.g., Guided Diffusion on Imagenet -> FFHQ fine-tuning) do not seem to be standard, and it is difficult to compare the results with the established baseline methods.
- It is difficult to understand where the efficiency gains of forward-pass running time come from. Is this mostly due to the “parameter scoring” (where some channels are pruned)? The authors describe using the gating approach to prune unnecessary activations, but in such cases, I do not believe the overall time for a forward pass should be reduced as the model still needs to perform a full forward pass (followed by additional gating at the end).
- The authors do not describe the limitations of the proposed approach (e.g., additional hyperparameters that need to be tuned or the additional model training required for distillation).

---

> ### Author Response · Authors · 2024-02-20
> **Reply to Reviewer FQW7's Review**
>
> Thank you very much for the thoughtful and detailed review. We reply point-by-point here, to begin the discussion. And we will submit a revised version of our paper soon.
>
> 1. **"Writing weakness"**: Certainly, we acknowledge the identified weaknesses in our writing. Allow us to elucidate the confusion mentioned and incorporate necessary revisions in the updated version of our paper.
>     - The sentence, "while, do users truly hunger for such an all-encompassing arsenal", means many users may not necessarily require the full range of functionalities offered by diffusion models. For instance, in the case of a pet store, the long-term need might solely revolve around the synthesis of images depicting cats and dogs.
>     - The phrase, "micro-managing model parameters", means we will conduct a detailed analysis of the parameters in the model, scrutinizing their significance at the channel level. This investigation will enhance our understanding of the model by delving into the nuanced impact of these parameters on its overall performance.
>     - The sentence, "Navigating the path to this optimal blend of efficiency and functionality is complex ...", means traditional parameter scoring methods are effective for single-step models but prove ineffective for multi-step diffusion models. We need to explore an approach that accurately assesses the importance of parameters in diffusion models, and does so within an acceptable timeframe.
>
>     We will address these perplexing writing issues in the revised version of our paper.
>
> 2. **"Experiments"**: Here, we will elucidate our experimental design for "Guided Diffusion on Imagenet -> FFHQ fine-tuning" you mentioned. The inspiration for this experiment's design is drawn from the Dream Booth. While we have confirmed that MoEDM performs well within its original training domain, our curiosity is piqued regarding its behavior when confronted with situations not encountered during pre-training. Intuitively, this represents a more challenging task. Therefore, we have incorporated this experiment into our study as a pivotal comparison against various baselines to assess MoEDM's performance in previously unseen conditions.
>
> 3. **"Faster forward"**: We sincerely apologize that our narrative has caused any confusion. In actuality, during the initial attempts and the parameter scoring, we simulated parameter removal by zeroing them out, as it was the simplest manipulation method. However, in the actual construction of MoEDM, we opted to completely delete the parameters, resulting in a substantial improvement in forward speed. The provided code robustly supports this approach. In MoEDM, the gating mechanism is only responsible for selecting a module to activate based on the current time step during sampling. We will highlight this aspect in the updated version of our paper to alleviate any reader confusion.
>
> 4. **"Limitations of MoEDM"**:
>     - First, MoEDM's performance is related to the image size employed during the diffusion process. We mentioned this issue in the "Subsets of ImageNet" paragraph of Section 4.3.1 of our paper, and we will further emphasize it in the updated version of our paper. Specifically, as outlined in Section 3.1 of our paper, the intermediate layers are relatively less crucial. Simultaneously, these intermediate layers handle smaller image sizes, while the outer layers handle larger ones, dictated by the UNet structure. If the original image size is excessively large, even with the removal of a substantial number of parameters from the intermediate layers, the computational load of the remaining layers on the sides remains significant. This could result in a performance decrease for MoEDM as the image resolution increases in the diffusion space, although the Latent Diffusion may alleviate this to a certain extent.
>     - Second, the training efficiency of MoEDM is still an area for improvement. For instance, different routing strategies can still be applied to the remaining layers. We have introduced this concept in the "Uneven Expansion" section of Section 4.3.2 in our paper, and we will further emphasize it in the updated version. Specifically, we propose assigning more expansion factors to the layers on the sides, as they are deemed more critical, and fewer expansion factors to the intermediate layers, given their relative importance. However, there is currently a lack of an automated allocation strategy.
>
> 5. **"Need for direct link to the Appendix"**: We greatly appreciate your suggestion to add a direct link to the Appendix, and we will make corrections in the updated version of our paper.
>
> 6. **"Hyperparameters"**: In this paper, the training hyperparameters employed are aligned with those specified for the corresponding tasks in the original diffusion models. It is important to note that they are not identical; we will emphasize this in the updated version of our paper.

---

### Review · Reviewer_LWwc · 2024-02-25

**Summary Of Contributions:**

This paper introduces a method introduces a new method for fine-tuning diffusion based image generation models.

This is achieved by discarding the middle layers of the U-net architecture of the model, and then creating a mixture of experts by replicating the model a number of times, and assigning each to some portion of the diffusion time in the diffusion model.

This is then trained by training this new model to match the outputs of the original model on a fine-tuning dataset.

This method is motivated by a desire on the authors part to reduce the sampling time of diffusion models. Long generation times, along with the large memory footprint of large diffusion models, is argued by the authors to be both barrier to widespread adoption of generative models, and also a waste of resources.

Experimentally, the authors on average a 50% speed up in sampling time, while memory footprint is reduced by approximately 20% (appendix C). This is achieved while maintaining similar, sometimes marginally better, sometimes marginally worse, performance than the original diffusion models, but only on the specific fine-tuning task.

**Audience:**

Yes

**Broader Impact Concerns:**

None.

**Claims And Evidence:**

Yes

**Requested Changes:**

Please engage with and make changes with respect to the weaknesses discussed above.

**Strengths And Weaknesses:**

The paper proposes an interesting idea to cut down the number of parameters in a diffusion model, and provides experimental evidence that cutting out the middle layers of a U-net is less disastrous than might be believed.

The proposed method and experiments achives the desired goal: a reduction in sampling time from diffusion models on class-specific generation tasks, with a limited class domain, and with little to no test time generalisation or domain-shift.

I have a number of weaknesses I would like discussed however.

Firstly, I find the motivation in the introduction unconvincing. The argument that authors present is that models that can generate a wide range of new and diverse images are unnecessary, and that a return to generating images in restricted classes is better for the average end user. I would argue this is untrue. For a reasonable period of time, the machine learning community has been able to generate reasonable quality images of restricted classes of images. These however were often very similar to those of the training data and not of great use in generating new and interesting images. It has only been in the last few years, where large scale models that can generate images across a large array of classes with some conditional prompting have existed, that these models have been of interest to the public, and indeed are now used regularly by the public and companies. Being able to create images, for example, of a dog in space riding a space ship along with its human friend in a cartoon style, is not something that it is demonstrated that distilled models from this method can achieve, but is exactly the kind of image a pet store may want to generate. The ability to generate cross-class and images far from the training data is exactly what makes large generative models interesting to end users.

On the side of increasing access to generative modeling, the cost (to the user) per image generated by a service such as Midjourey is on the order of 0.01 dollar, arguably not a prohibitive cost to the user. By contrast, having to define the class of images one is interested in, create a dataset of these images, fine-tune a model and then finally generate samples from it would seem likely to cost more than $0.005, given that the speedup of sampling from the method proposed is only 50%. One would need to find large numbers of users per fine-tuned model in order to make this sort of scheme viable. The existence and large utilisation of such services would also imply that cloud compute is both available enough, and stable enough, for such a service to exist.

Secondly, I found the description of the method a little confusing, and believe this could be improved. Section 3.1 is initially presented as the methodology employed to prune parameters in the model. As it turns out, this is not the case and it is only a piece of evidence used to motivate the whole layer pruning. A clear explanation of this would be less confusing. In section 3.2, the model is described in terms of gating of certain layers. While this is reminiscent of the mixture of experts set up, I think it is a little redundant given the gates are never trained. It could perhaps be clearer explained without reference to gates.

Thirdly, I found the language of the paper at times overstated, using overly complex or very strong languages, and often confusing due to word choice. This I think creates a unnecessary barrier to readership, particularly to those using English as a second language. Some examples, listed by page-paragraph-line:

1-2-1: "While," does not grammatically make sense to me. "truly hunger" to me is overly grand language.
1-2-2: the words "for a long time" are confusing here. Do you mean that the pet store may wish to generate such images over a long time period?
1-2-3: "egregious" is a very strong word.
2-1-3: The gammer construction either side of the citation does not follow. Perhaps a "however" is missing.
2-2-6: "At last" I think should be replaced with something to the effect of "Finally"
2-3-1: "Sampling velocity" is an unusual choice here, given it physical meaning of speed in a particular direction. Sampling speed may be more appropriate.
2-3-2: "drastic curtailment" might be better phrased as "significantly reduction"
3-2-2: "Unveil" may be better as "present"
4-1-4: "by their very nature, resonate with the ethos of" might be better stated as "fit into the framework of"
5-2-1: "effacious" -> effective
5-2-2: "For each channel in diffusion models" -> "for each channel in a diffusion model"
6-7-4: I am confused what "furnishing GPT3.5 with ..." means
6-7-5: I am confused why something is being promised in this line. I am not sure what this is supposed to mean.
8-2-2: I find the summary of the results confusing here. It is not clear which models are being compared at various points in the text, or what the conclusions drawn are. The phrase "to be undisputed" is particularly confusing.

---

> ### Author Response · Authors · 2024-02-26
> **Reply to Reviewer LWwc's Review**
>
> Thank you very much for the thoughtful and detailed review. We reply point-by-point here, to begin the discussion. And we will submit a revised version of our paper soon.
>
> 1. Of course, existing large-scale generative models do produce images across a wide range of categories. But this does not imply that in-depth exploration of image generation for specific categories becomes irrelevant. This is particularly crucial in certain application scenarios, such as medical imaging. Our MoEDM effectively integrates information from pre-trained models and applies it to specific tasks, achieving a noteworthy twofold increase in sampling speed.
>
>     Meanwhile, addressing the point you mentioned in "The ability to generate cross-class and images far from the training data is exactly what makes large generative models interesting to end users", relevant experiments have been conducted in the first paragraph of Section 4.3.1 of our paper. Detailed comparisons with baseline models were performed, substantiating the capability of MoEDM in generating images that deviate significantly from the training data.
>
>     For the cost of using generative models, it is true that the current cost is not high for users, but we believe that a model that can meet specific user needs in the long term and has significantly faster sampling speed will be welcomed by users.
>
>
>
>
>
> 2. Thank you for pointing out the confusion in the method description, especially the unclear explanation of Section 3.1 and the mention of gating strategy in Section 3.2. We will make modifications in the updated version of our paper to eliminate your confusion.
>
>
>
>
>
> 3. Thank you for pointing out the vocabulary usage issues in our paper. We will explain them to you one by one and make revisions in the updated version of our paper:
>     - "1-2-1": We will modify it to "Do users truly need such a comprehensive but huge model?" in the updated version of our paper.
>     - "1-2-2": This sentence suggests that pet stores will consistently maintain these specific requirements for an extended period, rather than frequently changing them in the short term. This trend is applicable to many industries. We will clarify this point in the updated version of our paper.
>     - "1-2-3": We will modify the "egregiously" to "severely" in the updated version of our paper.
>     - "2-1-3": The "however" is indeed missing, and we will add it in the updated version of our paper.
>     - "2-2-6": Indeed, the "Finally" will be a better choice, and we will modify it in the updated version of our paper.
>     - "2-3-1": The meaning of "velocity" does not fit the current context, and we will modify it to "speed" in the updated version of our paper.
>     - "2-3-2": Indeed, the "significant reduction" is a better choice, and we will modify it in the updated version of our paper.
>     - "3-2-2": Indeed, the "present" is a better choice, and we will modify it in the updated version of our paper.
>     - "4-1-4": Indeed, the "fit into the framework of"  is a better choice, and we will modify it in the updated version of our paper.
>     - "5-2-1": The "efficacious" should indeed be replaced with "effective" in the updated version of our paper.
>     - "5-2-2": Here, "in a diffusion model" is indeed more appropriate, and we will modify it in the updated version of our paper.
>     - "6-7-4" and "6-7-5": The purpose of this paragraph is to illustrate how we utilized the help of GPT3.5 when collecting data for distillation. And we have confirmed the randomness of the data to ensure that there will be no random seeds and text prompts that are the same as the sampling stage. We will further clarify them in the updated version of our paper.
>     - "8-2-2": The ablation experiment in this paper mainly consists of a comparison between "MoEDM w/o Dynamic" and MoEDM. Another type of ablation, "using a full-size model to construct dynamic routing," is equivalent to fine-tuning the full-size model without fundamentally contributing to the improvement of sampling speed, which is obvious. That's why we used the phrase "to be undisputed".
>
>     We will soon make corrections to the issues mentioned above in the updated version of our paper.

---

### Decision · Action_Editor_nGkM · 2024-07-15

**Recommendation:** Reject

**Comment:**

Reviewers find that the paper isn't ready for publication mainly due to lack of clarity about the proposed method and the writing quality of the paper. Another point raised in the reviews that the authors may want to think about and expand upon in the paper is how the proposed method would be used in practice and solve the problem of cheaper generation with diffusion models (that it claims to solve), since the cost of curation of the finetuning dataset could be a major task.

**Audience:**

The topic of the paper is relevant to the TMLR audience - improving the sampling time of diffusion is an important problem. However the findings of the paper may have rather limited appeal in AE's view due to the need of creating a rather large finetuning dataset (~1k images) to create the MoE model which yields 50% improvement in sampling time (as also highlighted by a reviewer).

**Claims And Evidence:**

The paper proposes an approach for finetuning a pretrained general purpose diffusion model to a narrow domain targeting efficiency in sampling. It finds that a subset of the middle layers of the UNet can be pruned to achieve this. It creates a mixture of experts by replicating the pruned model a number of times, assigning each expert to some portion of the diffusion time in the diffusion process. The MoE is trained to match the outputs of the original model on a fine-tuning dataset. The paper achieves on average a 50% speed up in sampling time, while maintaining the quality of generation on the narrower domain.

Reviewers find the paper lacking in mainly following two aspects:

- Lack of clarity about the approach: this point came up in the post-rebuttal assessment from the reviewers. It is not clear how Eq. (1) is used to prune the channels since the objective has randomness due to noise in the diffusion process. Does it need an expectation over the the samples and noise? The notion of discard ratio also remains unclear after the author response and how it affects the pruning strategy. Finally, Eq. (1) is presented for pruning channels, but then layers are removed. It remains unclear why this is and its motivation. A clear presentation of the pruning strategy is needed: perhaps an Algorithm table with clear steps will be useful here.

- Writing quality: This is shared concern by all reviewers that writing clarity of the paper can be substantially improved. The key ideas of the paper are burdened by unnecessary use of ornamental language. Authors should take reviewers' comments into account on this and consider revising the paper to make it easier to read and get the main ideas of the proposed method.

**Resubmission Of Major Revision:**

The authors may consider submitting a major revision at a later time.